# Bond Properties of Magnesium Phosphate Cement-Based Engineered Cementitious Composite with Ordinary Concrete

**DOI:** 10.3390/ma15144851

**Published:** 2022-07-12

**Authors:** Hongtao Chang, Hu Feng, Zeyu Guo, Aofei Guo, Yongkang Wang

**Affiliations:** 1School of Civil Engineering, Zhengzhou University, Zhengzhou 450001, China; changhongtao@zzmetro.cn (H.C.); fenghu@zzu.edu.cn (H.F.); 18848963830@163.com (Z.G.); wangyongkang1201@126.com (Y.W.); 2Zhengzhou Metro Group Co., Ltd., Zhengzhou 450001, China

**Keywords:** polyvinyl alcohol fibers, magnesium phosphate cement, engineered cementitious composite, shear bond strength, bond-slip model

## Abstract

A magnesium phosphate cement-based engineered cementitious composite (MPC-ECC) was developed using polyvinyl alcohol (PVA) fibers and fly ash. In this study, the bond behavior of MPC-ECC with ordinary concrete was evaluated through single and double shear bond strength tests. The effects of the water to solid mass ratio (W/S), the sand to binder mass ratio (S/B), the molar ratio of MgO to KH2PO4 (M/P), the fly ash content (F), the borax dosage (B), the volume fraction of PVA fibers (Vf), and curing age on the bond behavior of MPC-ECC with ordinary concrete were examined. The results showed that as the W/S increased, the single and double shear bond strengths were gradually reduced. As the S/B increased, the double shear bond strength increased; the single shear bond strength first decreased up to an S/B of 0.1 and then increased. With the increase of M/P, the single and double shear bond strengths increased. With the increase of F, the single shear bond strength first increased up to an F of 30% and then decreased; the double shear bond strength decreased. With the increase of B, the single and double shear bond strengths increased first and then decreased, and their strength reached its maximum at a B of 6%. The increase of Vf improved the single and double shear bond strengths. The research results can provide some technical guidance for repairing concrete structures with MPC-ECC.

## 1. Introduction

Concrete is currently the most used building material in the world, and a significant number of concrete structures need to be repaired or strengthened every year due to adverse weather conditions and heavy mechanical loads [1]. This process requires a large amount of material and financial resources. Faced with the global problem of serious degradation of concrete buildings, it is of great importance to select appropriate materials and methods for repairing and strengthening.

When using ordinary portland cement (OPC) as the repairing material, repair operations can disrupt the use of structures for a long time. However, in some special cases, the structure needs to be repaired quickly and then put into use immediately, so rapid-setting cementing materials are needed. Magnesium Phosphate cement (MPC), a kind of inorganic cementitious material, is made of dead-burned magnesium oxide, acid phosphate, and a certain amount of borax as a retarder [1]. MPC has several advantages [2,3,4,5,6,7,8,9], such as high early strength, excellent durability, superior adhesive strength, low permeability, rapid hardening, and good compatibility with ordinary concrete. MPC gains strength over a short duration as compared to several hours with OPC [9,10]. Yang et al. [11,12] indicated that MPC mortar has lower shrinkage than OPC mortar, which made the MPC mortar show better volume stability than the OPC mortar, and the coefficient of thermal expansion of MPC mortar was very close to that of OPC mortar, which was a prerequisite for repairing OPC-based composites using MPC-based composites. In addition, compared with OPC, the production of MPC has much lower carbon dioxide emissions. Therefore, MPC is better than OPC as a repairing material for concrete structures.

It is well known that in addition to the intrinsic behavior of MPC-based composites, their bond performance with ordinary concrete is a key index to the repair quality [13,14]. Qiao et al. [8] stated that the bond strength of MPC mortar with ordinary concrete was much higher than that of OPC mortar with ordinary concrete in various stress conditions. Jiang et al. [15] found that the 7-day bending strength of MPC pastes bonded with ordinary concrete was up to 6.2 MPa, and the MPC paste showed good compatibility with the ordinary concrete. In addition, Yang et al. [12,16] indicated that when using MPC to repair damaged concrete, it is not necessary to wet the surface of the concrete, which indicated that MPC mortar can be conveniently used. Therefore, MPC-based composites can be considered promising materials for ordinary concrete structures.

In today’s engineering applications, repair materials are required to have not only high strength and durability but also good ductility to ensure engineering quality and safety. For MPC, its brittle and lower strain hardening properties need to be solved [6,17,18] to expand its application range. Victor C. Li et al. [19] first introduced the concept of engineered cementitious composites (ECC) in the early 1990s. Generally, the ECC shows strain hardening behavior, with the ultimate tensile strain exceeding 3%. In recent decades, more and more scholars have been using low volume fractions of randomly distributed short fibers to develop ECC, such as polyvinyl alcohol fiber (PVA) [20,21,22,23], polyethylene fiber (PE) [24,25,26,27,28], polypropylene fiber (PP) [29], and steel fiber (SF) [30,31]. It is worth mentioning that ECC has a strain energy density of up to 1000 kJ∙m^−3^ [32]. In our group’s previous research [33], PVA fibers were incorporated into MPC-based composites to produce magnesium phosphate cement-based engineered cementitious composite (MPC-ECC) with deflection hardening behavior and high ductility to address the brittleness of MPC-based composites. However, there is no study on the bond properties of MPC-ECC with ordinary concrete, so it is necessary to conduct such a study for the application of MPC-ECC as a repair material.

In this study, MPC-ECC was prepared using MPC and PVA fibers, and fly ash (FA) was added to improve the hydration degree and pore structure of MPC [34,35]. The effects of the water to solid mass ratio (W/S), the sand to binder mass ratio (S/B), the molar ratio of MgO to KH2PO4 (M/P), the fraction of fly ash (F), the borax dosage (B), the volume fraction of PVA fibers (Vf), and curing age on the bond behavior of MPC-ECC with ordinary concrete were examined through single and double shear bond strength tests. All these variables were selected to establish the bond–slip model between MPC-ECC and concrete. This study provides some theoretical support for the application of MPC-ECC in the field of structural rehabilitation of buildings.

## 2. Experimental Program

### 2.1. Materials

Highly ductility magnesium phosphate cement-based engineered cementitious composites (MPC-ECC) were prepared from calcined magnesium oxide (MgO), potassium dihydrogen phosphate (KH_2_PO_4_), borax, quartz sand, fly ash (FA), and polyvinyl alcohol (PVA) fibers. The MgO (purity: ≥96.5%) was sourced from Huanai Magnesium Industry Co., Ltd. of Yancheng, Jiangsu, China, and then ground using a ball mill (SM500 × 500, Beijing Zhongke Road Construction Equipment Co., Ltd.) to achieve a specific surface area of 359.8 m^2^/kg. The KH_2_PO_4_ (white crystalline powder, purity: 99.5%) with two different degrees of fineness (0.08–0.630 mm and 0.150–1.180 mm) were purchased from Tianjin Dingshengxin Chemical Co. of China and Henan Huaxing Chemical Co. of China, respectively. The KH_2_PO_4_ used in this study was prepared by mixing the two types of KH_2_PO_4_ at a mass ratio of 1:1 and then passing them through a sieve to obtain a better particle size distribution as shown in Figure 1. Quartz sand with two particle size ranges of 0.125–0.180 mm and 0.090–0.100 mm was purchased from Henan Zhongbang Environmental Protection Technology Co. Ltd. of China. Similarly, to obtain a good particle size distribution, these two types of quartz sand were mixed at a mass ratio of 1:1, with the technical specifications shown in Table 1. The FA used was first-class powered coal ash sourced from Henan Kanghui Cement Products Co. of China, with the technical specifications and the chemical composition shown in Table 2 and Table 3, respectively. The particle gradation curves of MgO, FA, and quartz sand are shown in Figure 2. In this study, the borax (Na_2_B_4_O_7_·H_2_O, purity: 99.5%) was used as a retarder, purchased from Liaoning Borda Technology Co. of China. The PVA fibers were produced from Kuraray, Japan, with the performance indicators presented in Table 4.

Ordinary portland cement, sand, and graded gravel were used as raw materials to prepare ordinary concrete (Grade C40 defined by GB/T 50107-2010 [36], the most common strength class used in construction in China) in the study. P.O.42.5 ordinary portland cement (specific surface area: 363 m^2^/kg) defined by GB 175-2007 [37] was supplied by Zhengzhou Duoyoubang Technology Co. of China, with the chemical composition and the mineral composition shown in Table 5 and Table 6, respectively. The compressive strength and flexural strength of the cement after 3 days are 27.6MPa and 6.0 MPa, respectively. River sand with a fineness modulus of 2.3–3.0 and a particle size range of 0.35–0.5 mm was used as fine aggregate. Graded gravel with a particle size range of 1–2 cm was used as coarse aggregate.

### 2.2. Mixture Proportions

According to our group’s previous research, the compression and bending properties of MPC-ECC are better when the sand to binder (MgO, KH_2_PO_4_, and FA) mass ratio (*S*/*B*) is 0.2, the water to solid (MgO, KH_2_PO_4_, FA, and quartz sand) mass ratio (*W*/*S*) is 0.13, the retarder borax dosage (*B*) is 6% by mass of MgO, the volume fraction of PVA fibers (*V*_f_) is 2.0%, the molar ratio of MgO to KH_2_PO_4_ (*M*/*P*) is 4, and the FA content (*F*) is 30% by mass of MgO and KH_2_PO_4_ [38]. Therefore, this mixing proportion was used as the base MPC-ECC group in this study. The effects of *W*/*S* (0.10, 0.13, and 0.16), *S*/*B* (0, 0.1, and 0.2), *M*/*P* (3, 4, and 5), *B* (3%, 6%, and 9%), *V*_f_ (1.2%, 1.6%, and 2.0%), and *F* (20%, 30%, and 40%) on the interfacial bond properties of MPC-ECC with ordinary concrete (C40) were investigated. The mixing proportions are shown in Table 7. Meanwhile, based on the base group, the effects of curing age (3 days, 7 days, and 28 days) of MPC-ECC on its bond properties with ordinary concrete were examined. Lastly, the bond properties of ordinary concrete (C40) with ordinary concrete (C40), denoted as the control group, were also examined for comparison with those of MPC-ECC with ordinary concrete (C40).

### 2.3. Specimen Preparation

In this study, the specimens for bond tests were prepared in two steps: the first step was to prepare the ordinary concrete specimen; the MPC-ECC specimen was prepared in the second step, as explained in detail below.

(1) Preparation of the ordinary concrete specimen

A single horizontal shaft concrete mixer (Type: HX-15, Cangzhou Zhongke Beigong Testing Instrument Co., Ltd., Cangzhou, China) was used to mix the raw materials for ordinary concrete. To avoid excessive water loss during the mixing process, the blades and the inner surface of the mixer were moistened. Firstly, the weighed cement, sand, and graded gravel were added to the mixer in turn and mixed for 120 s to make a homogeneous mixture; secondly, water was added and mixed again for 120 s; lastly, the mixture was cast in the mold and then vibrated for 60 s on the vibration table. The mold could be dismantled after 24 h of environmental curing in the laboratory. The ordinary concrete specimens were then placed in a standard curing room (temperature: 20 ± 2 °C, relative humidity: >90%) for 14 d. After that, the specimens were sprayed with a high-pressure water gun for five minutes to simulate the damaged concrete surface morphology and then placed back in the standard curing room for another 14 d.

(2) Preparation of the MPC-ECC specimen

The weighed MgO, KH_2_PO_4_, quartz sand, borax, and fly ash were first added to a bucket in turn, and a hand-held high-powered electric mixer was used to dry mix the raw materials for 60 s to achieve a homogeneous mixture. This was followed by a further 120 s of mixing with water, during which the fibers were added and then mixed for another 60 s. The mixture was poured into a customized mold in which the ordinary concrete specimen with the surface treatment was placed in advance. The mixture was firstly vibrated for 60 s and then vibrated for 30 s after an interval of 15 s on the vibration table. The mold could be dismantled after 2 h of environmental curing in the laboratory to obtain the MPC-ECC specimens bonded with ordinary concrete. The specimens were then placed in the standard curing chamber (temperature: 20 ± 2 °C, relative humidity: >90%) and cured until the test age (3 days, 7 days, and 28 days). Three specimens were prepared for each mix proportion.

The control group (ordinary concrete bonded with ordinary concrete), as mentioned in Section 2.2, were then cast for comparison analysis. The control specimens were prepared in a similar way to preparing MPC-ECC specimens bonded with ordinary concrete except that the MPC-ECC was replaced with ordinary concrete and then cured in the standard curing room (temperature: 20 ± 2 °C, relative humidity: >90%) for 28 days.

### 2.4. Test Methods

The single and double shear bond strength tests were conducted to examine the bond properties of MPC-ECC with ordinary concrete, as introduced below. It is noted that for each test herein, the bond properties of the control group (between ordinary concretes) were also examined for comparison analysis.

#### 2.4.1. Single Shear Bond Strength Test

For testing the shear resistance of MPC-ECC with ordinary concrete, the commonly used method is the single shear bond strength test. The shear specimen was a 100 mm × 100 mm × 100 mm cube, which was composed of ordinary concrete and MPC-ECC. The size of both the ordinary concrete specimen and the MPC-ECC specimen was 100 mm × 100 mm × 50 mm and three specimens were tested for each mix proportion, as shown in Figure 3. The electronic universal testing machine (model: WDW-100, specification: 100 kN) was used with a loading rate of 0.25 mm/min. The load and slip were simultaneously collected by dynamic signal acquisition and analysis system, and then the bond–slip curve was drawn. The shear strength equation for the interface between the MPC-ECC and the ordinary concrete is shown in Equation (1).
*τ*_s_ = *F*_s_/*A*_s_(1)
where *τ*_s_ is the single shear bond strength (MPa); *F*_s_ is the axial ultimate shear load (kN); *A*_s_ is the area of bond surface, 100 mm × 100 mm.

#### 2.4.2. Double Shear Bond Strength Test

The double shear bond strength test was carried out according to the standard JGJ/T 221-2010 [39], as illustrated in Figure 4. The electronic universal testing machine (model: WDW-100, specification: 100 kN) was used in this test. The specimen for this test was a 100 mm × 100 mm × 300 mm cuboid, with ordinary concrete in the middle and MPC-ECC on both sides. The specimen size of both ordinary concrete and MPC-ECC was 100 mm × 100 mm × 100 mm, and three specimens were tested for each mix proportion. Before loading, the pedestal should only be in contact with the MPC-ECC, and the upper loading plate should only be in contact with the ordinary concrete surface. The loading of the electronic universal testing machine was controlled at a rate of 0.25 mm/min until the specimen was damaged and the ultimate load recorded. The formula for calculating the shear strength is shown in Equation (2).
*τ*_d_ = *F*_d_/*A*_d_(2)
where *τ*_d_ is the double shear bond strength (MPa); *F*_d_ is the ultimate load, kN; *A*_d_ is the area of bond surface, 2 × 100 mm × 100 mm for this test.

## 3. Results and Discussion

### 3.1. Effect of Water to Solid Mass Ratio (W/S)

The single and double shear bond strength values of ordinary concrete with MPC-ECC at *W*/*S* of 0.10, 0.13, and 0.16 are shown in Figure 5. It can be seen that there is a negative correlation between the *W*/*S* and the shear bond strength. This is because as *W*/*S* decreases, the amount of pores between hydration products decreases, and the MPC-ECC becomes denser [40]. As a result, the MPC-ECC adheres tightly to the ordinary concrete, increasing the shear bond strength when the *W*/*S* is small. It has been previously reported that lower *W*/*S* makes porosity lower and the specimens denser [41]. On the contrary, when *W*/*S* is high, the excess water evaporates from the voids, and the specimens shrink and deform, which can accelerate the development of cracks at the bond surface and therefore reduce the bond strength. Meanwhile, the shear bond strengths of MPC-ECC with ordinary concrete at different *W*/*S* are higher than that of the control group. It can be concluded that the shear resistance of MPC-ECC with ordinary concrete at different *W*/*S* is better than that of the control group.

### 3.2. Effect of Sand to Binder Mass Ratio (S/B)

The single and double shear bond strength values of ordinary concrete with MPC-ECC at *S*/*B* of 0, 0.1, and 0.2 are shown in Figure 6. The shear strength varies with the change of *S*/*B*. The single and double shear bond strengths reach their maximum values at *S*/*B* of 0 and 0.2, respectively. However, it can be seen from the test results that there is little difference in the single or double shear bond strength at different *S*/*B*. In other words, *S*/*B* has little influence on the shear bond strength of MPC-ECC with ordinary concrete. In addition, the single and double shear bond strengths of the specimens at different *S*/*B* are higher than those of the control group.

### 3.3. Effect of Magnesia to Phosphate Molar Ratio (M/P)

The single and double shear bond strength values of ordinary concrete with MPC-ECC at *M*/*P* of 3, 4, and 5 are shown in Figure 7. It can be seen that the increase of *M*/*P* can improve the single and double bond strengths. This is probably because, as the *M*/*P* increases, the hydration products could grow much larger and turn into bladed and prismatic crystals with wrinkled surfaces [42]. The larger hydration products might be beneficial to the development of the compressive strength of specimens. What matters is that there is a positive correlation between the shear bond strength and the compressive strength i.e., high compressive strength of the MPC-ECC is associated with its high shear bond strength with ordinary concrete [43,44]. Due to the unevenness of the ordinary concrete surface, after the MPC-ECC was cast on top of it, both the MPC-ECC and the ordinary concrete had their own parts embedded in one another, and the MPC-ECC embedded in concrete generated principal compressive stresses when loading. When the principal compressive stress exceeded the compressive strength of MPC-ECC, cracks were created, and the MPC-ECC/ordinary concrete specimens were eventually broken down [44]. Wang et al. indicated that within the *M*/*P* range of 3–5, the compressive strength of MPC mortar was increased [45]. The same pattern was shown by Yang. et al. [46]. In general, as the *M*/*P* increases, the compressive strength of the MPC-ECC is increased, and the appearance of cracks is delayed, thus increasing the shear bond strength of ordinary concrete with MPC-ECC.

### 3.4. Effect of FA Content (F)

The single and double shear bond strength values of ordinary concrete with MPC-ECC at *F* of 20%, 30%, and 40% are shown in Figure 8. It can be seen that the shear bond strengths are relatively high at *F* of 20% and 30%. The reason for this may be the morphological effect [47] of fly ash on the MPC-FA system in that the smooth, spherical particles can improve the performance of slurry [46,48], which facilitates a tighter bond between MPC-ECC and ordinary concrete and improves the shear bond strength. However, when *F* is too large (40% in this study), the shear bond strengths are reduced, because the excessive FA substitution reduces the amount of cementitious material, thus reducing the amount of hydration products in the specimen.

### 3.5. Effect of Borax Content (B)

The single and double shear bond strength values of ordinary concrete with MPC-ECC at *B* of 3%, 6%, and 9% are shown in Figure 9. It can be seen that 6% borax content results in the most significant improvement in the single and double shear bond strengths. This is because the retarder borax can increase the flowability and permeability of the slurry when *B* = 6%, which allows MPC-ECC slurry to penetrate the pores of the concrete surface more easily, thus improving the bond strength. However, as the borax content continues to increase, there exists excess borax not involved in the reaction. The borax crystals have a smooth surface and low strength, so they cannot effectively adhere to other hydrates in the hardened MPC and become weak points on the bond surface, reducing the shear bond strength [49]. Therefore, the retarder borax content should be strictly controlled and is suggested to be 6% in this study.

### 3.6. Effect of Fiber Volume Fraction (V_f_)

The single and double shear bond strength values of ordinary concrete with MPC-ECC at *V_f_* of 1.2%, 1.6%, and 2.0% are shown in Figure 10. The increase of fiber volume fraction increases the shear bond strength, but only to a limited extent, which is probably because the higher fiber volume fraction may increase the bridging action at the bond surface, resulting in higher shear bond strength. Furthermore, the incorporation of PVA fiber delays the propagation of cracks that causes structural failure, as their higher modulus of elasticity allows them to consume more energy during deformation, which helps resist cracking.

### 3.7. Effect of Curing Age

The single and double shear bond strength values of ordinary concrete with MPC-ECC at curing ages of 3d, 7d, and 28d are shown in Figure 11. It can be seen that the single shear bond strength increases with curing age because the amount of hydration products gradually increases so that the strength of MPC-ECC itself and the bond between MPC-ECC and ordinary concrete continue to develop. The single shear bond strengths at 3d, 7d, and 28d can reach 82.80%, 93.6%, and 197.5% of that of the control group at 28d, respectively. The double shear bond strength also shows the same trend as the single shear bond strength. It can be concluded that the bond property of MPC-ECC with ordinary concrete is similar to that of the control group at 28d in the early stages, and the shear bond strength increases further with the increase of curing age.

## 4. Bond–Slip Model of MPC-ECC with Ordinary Concrete

In this study, a calculation model of single shear bond strength considering fiber volume fraction, borax content, *M*/*P*, *W*/*S*, *S*/*B*, and fly ash content of the MCC-ECC matrix was proposed. Bond–slip relationship curves obtained under different variables are shown in Figure 12.

Based on the above results, this study chose the ultimate shear bond stress, the slip value corresponding to the ultimate shear bond stress, and the interfacial fracture energy as eigenvalues, and proposed a simplified bond–slip constitutive relationship model, as shown in Equations (3)–(7) [2].
(3)$$\left\{ \begin{array}{l} \tau_{c}=\tau_{u}\frac{S}{S_{u}}\;\;0\leq S\leq S_{u}\\ \tau_{c}=\tau_{u}\frac{S_{f}-S}{S_{f}-S_{u}}\;S_{u}<S\leq S_{f}\\ \tau_{c}=0\;\;\;\;S>S_{f} \end{array}\right.$$
*τ_u_* = *ατ*_*u*1_*τ*_*u*2_*τ*_*u*3_*τ*_*u*4_*τ*_*u*5_*τ*_*u*6_(4)
*S_u_* = *βS*_*u*1_*S*_*u*2_*S*_*u*3_*S*_*u*4_*S*_*u*5_*S*_*u*6_(5)
*G_f_* = *γG*_*f*1_*G*_*f*2_*G*_*f*3_*G*_*f*4_*G*_*f*5_*G*_*f*6_(6)
(7)$$S_{f}=\frac{2G_{f}}{\tau_{u}}$$
where *τ_c_* is the calculated shear bond stress; *τ_u_* is the ultimate shear bond stress; *S_u_* is the slip value corresponding to the ultimate shear bond stress; *G_f_* is the interfacial fracture energy; *S_f_* is the ultimate slip value; *α*, *β*, and *γ* are fitting coefficients; *τ_u_*_1_, *S_u_*_1_, and *G_f_*_1_ are influence coefficients of fiber volume fraction on *τ_u_*, *S_u_*, and *G_f_*, respectively; *τ_u_*_2_, *S_u_*_2_, and *G_f_*_2_ are influence coefficients of borax content on *τ_u_*, *S_u_*, and *G_f_*, respectively; *τ_u_*_3_, *S_u_*_3_, and *G_f_*_3_ are influence coefficients of *M*/*P* on *τ_u_*, *S_u_*, and *G_f_*, respectively; *τ_u_*_4_, *S_u_*_4_, and *G_f_*_4_ are influence coefficients of *W*/*S* on *τ_u_*, *S_u_*, and *G_f_*, respectively; *τ_u_*_5_, *S_u_*_5_, and *G_f_*_5_ are influence coefficients of *S*/*B* on *τ_u_*, *S_u_*, and *G_f_*, respectively; *τ_u_*_6_, *S_u_*_6_, and *G_f_*_6_ are influence coefficients of fly ash content on *τ_u_*, *S_u_*, and *G_f_*, respectively. The following (Figure 13, Figure 14 and Figure 15) are the results of polynomial fitting for each influence coefficient.
*τ_u_* = 0.0035(3.1 − 1.125*V_f_* + 0.5625*V_f_*^2^)(0.09 + 0.865*B* − 0.06056*B*^2^)[−7.68 + 4.755(*M*/*P*) − 0.515 (*M*/*P*)^2^] [−0.65556 + 90.27778(*W*/*S*) − 472.2222 (*W*/*S*)^2^][3.3 − 4.4 (*S*/*B*) + 17 (*S*/*B*)^2^] (0.01 + 0.238*F* − 0.0045*F*^2^)(8)
*S_u_* = 153.73644 (0.12737 + 0.11862*V_f_*) (0.1223 + 0.06812*B* − 0.0046*B*^2^)[−0.5838 + 0.47486 (*M*/*P*) − 0.05939 (*M*/*P*)^2^] [−1.07211 + 25.30722 (*W*/*S*) − 109.61111 (*W*/*S*)^2^][0.3225−0.1785(*S*/*B*)+1.965(*S*/*B*)^2^](0.1203+0.01233*F*−0.0001385*F*^2^)(9)
*G_f_* = 6.46948 (0.6315 − 0.07438*V_f_* + 0.05219*V_f_*^2^) (−0.3888 + 0.33868*B* − 0.02644*B*^2^)[−3.5933 + 1.9944 (*M*/*P*) − 0.2308 (*M*/*P*)^2^] [−3.81416 + 75.78944 (*W*/*S*) − 316.38889 (*W*/*S*)^2^][0.6071 − 3.314 (*S*/*B*) + 18.36 (*S*/*B*)^2^] (−1.1457 + 0.12322*F* − 0.00207*F*^2^)(10)

The fitting parameters *α* = 0.0035, *β* = 153.73644, and *γ* = 6.46948 are obtained by nonlinear curve fitting. Equations (4)–(6) can be written in the form of Equations (8)–(10).

## 5. Conclusions

This paper investigates the effects of material proportion (water-solid mass ratio, sand to binder mass ratio, molar ratio of MgO to KH_2_PO_4_, FA content, borax content, and fiber volume fraction) and curing age on the single and double shear bond strengths between MPC-ECC and ordinary concrete. The bond properties of MPC-ECC are better when the *W/S* is 0.10, *M*/*P* is 5, *B* is 6%, *V_f_* is 2%, *F* is in the range of 20%–30%, and *S*/*B* has little effect on its bond performance:(1)The shear bond strength decreases as the *W*/*S* increases. The single shear bond strength and double shear bond strength are the highest at the *W*/*S* of 0.10, reaching 3.65 MPa and 1.99 MPa, respectively. But as the *S*/*B* increases, the effects of *S*/*B* on single shear bond strength and double shear bond strength are not obvious.(2)As the *M*/*P* and the fiber volume fraction increase, the shear bond strength increases. The single and double shear bond strengths are highest at *M*/*P* = 5, 3.22 MPa, and 1.85 MPa, respectively. This result is probably due to the microscopic morphology of hydration products at different *M*/*P*. The shear bond strength is highest at a fiber volume fraction of 2.0%.(3)The single and double shear bond strengths are highest at 30% and 20% FA content, 3.10 MPa and 1.70 MPa, respectively. The bond effect is best when FA content is between 20% and 30%, while the development of bond strength is not good when FA content is too high. Similarly, the single and double shear bond strengths of the specimens are highest at a borax dosage of 6%, reaching 3.10 MPa and 1.61 MPa, respectively. A too high or too low borax dosage is not conducive to the development of the bond strength.(4)The single and double shear bond strengths after 3 d of standard curing of MPC-ECC reached nearly 82.8% and 66.7% of those of the control group at 28 d, respectively. The double shear bond strength at 7 d reaches 100% of that of the control group at 28 d. The single and double shear bond strengths after 28 d of standard curing are all higher than those of the control group at 28 d, nearly twice.

## Figures and Tables

**Figure 1 materials-15-04851-f001:**
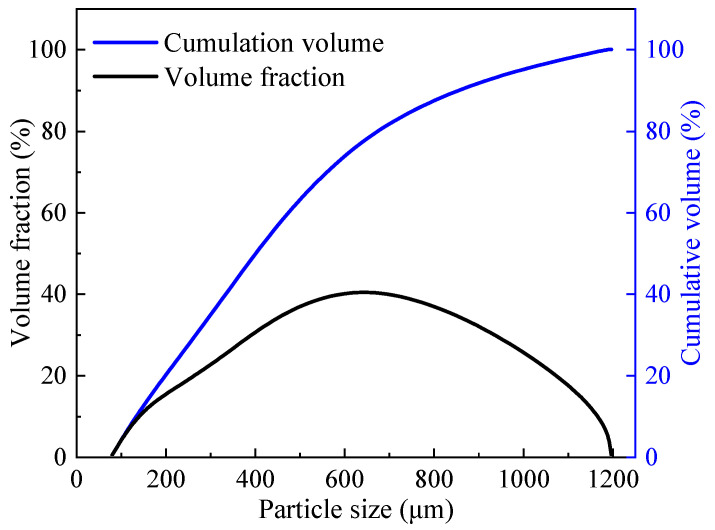
Particle size distribution of KH_2_PO_4._

**Figure 2 materials-15-04851-f002:**
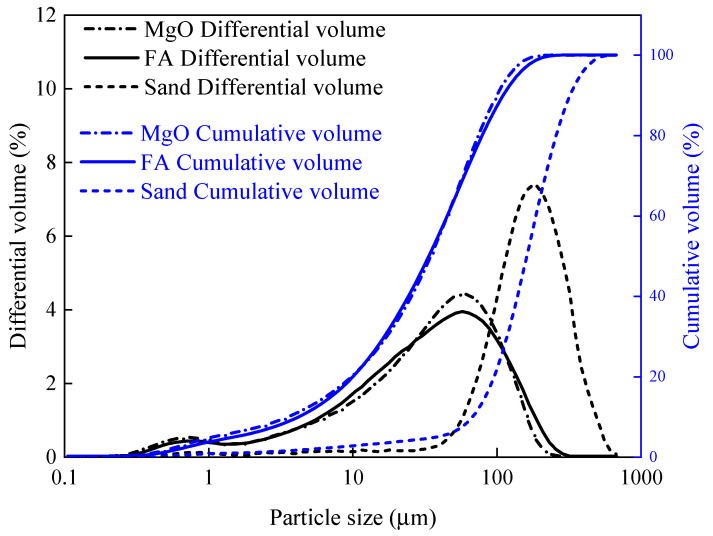
Particle size distribution of MgO, FA, and quartz sand.

**Figure 3 materials-15-04851-f003:**
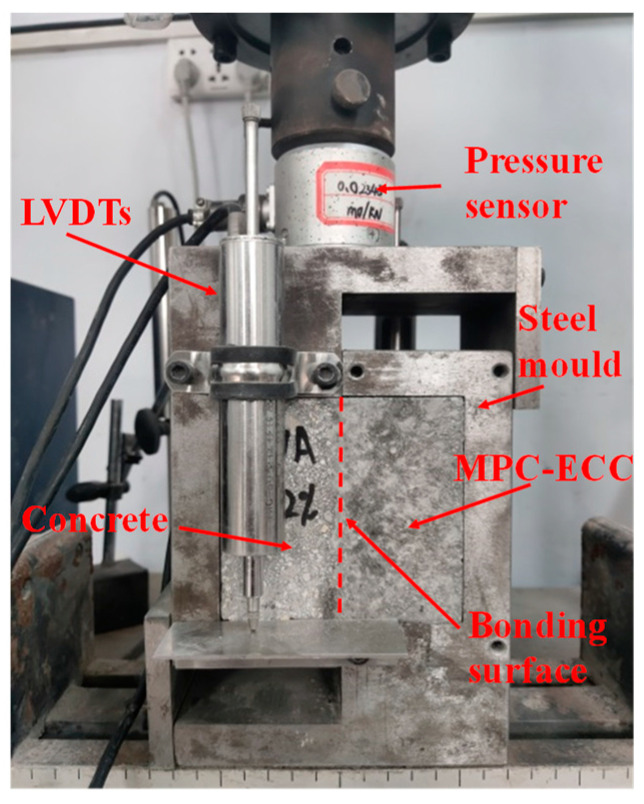
Setup of the single shear bond strength test.

**Figure 4 materials-15-04851-f004:**
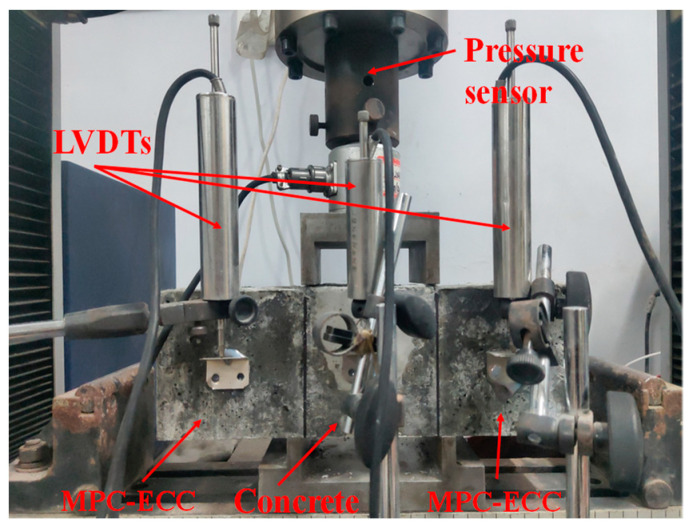
Setup of the double shear bond strength test.

**Figure 5 materials-15-04851-f005:**
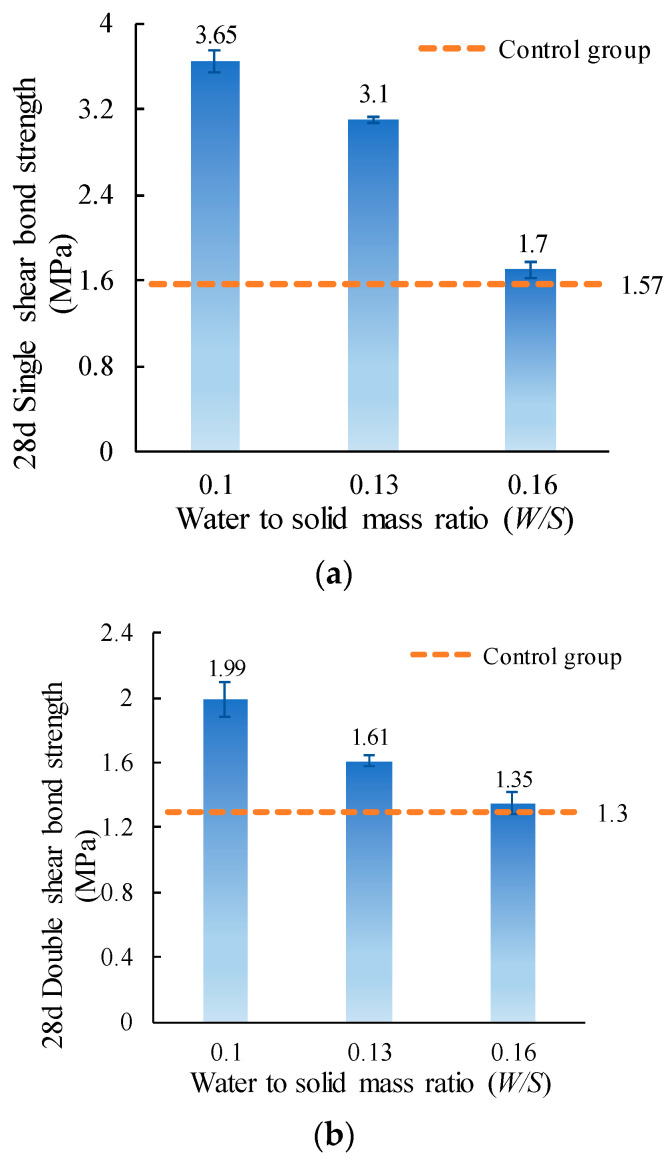
(**a**) Single shear bond strength and (**b**) double shear bond strength of specimens with varying *W*/*S*.

**Figure 6 materials-15-04851-f006:**
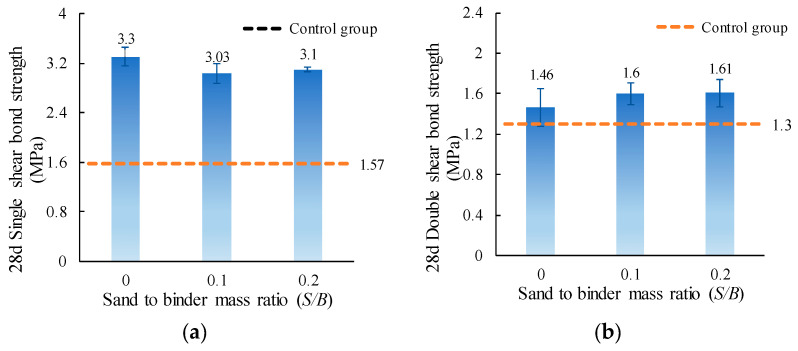
(**a**) Single shear bond strength and (**b**) double shear bond strength of specimens with varying *S*/*B*.

**Figure 7 materials-15-04851-f007:**
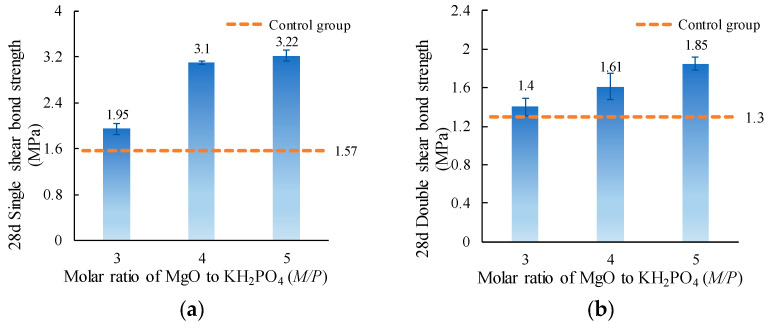
(**a**) Single shear bond strength and (**b**) double shear bond strength of specimens with varying *M*/*P*.

**Figure 8 materials-15-04851-f008:**
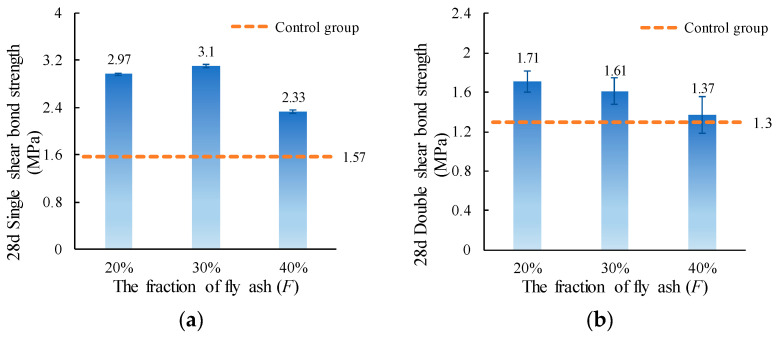
(**a**) Single shear bond strength and (**b**) double shear bond strength of specimens with varying *F*.

**Figure 9 materials-15-04851-f009:**
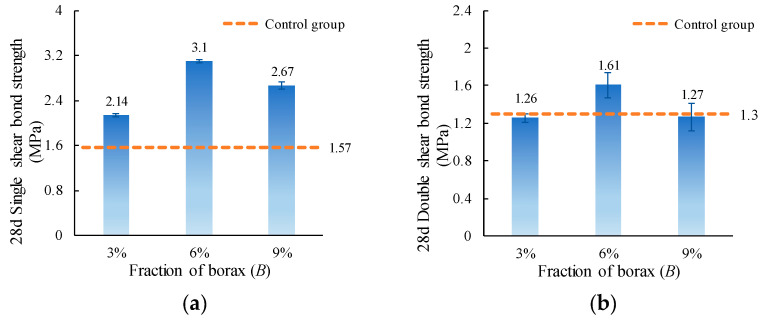
(**a**) Single shear bond strength and (**b**) double shear bond strength of specimens with varying borax content (*B*).

**Figure 10 materials-15-04851-f010:**
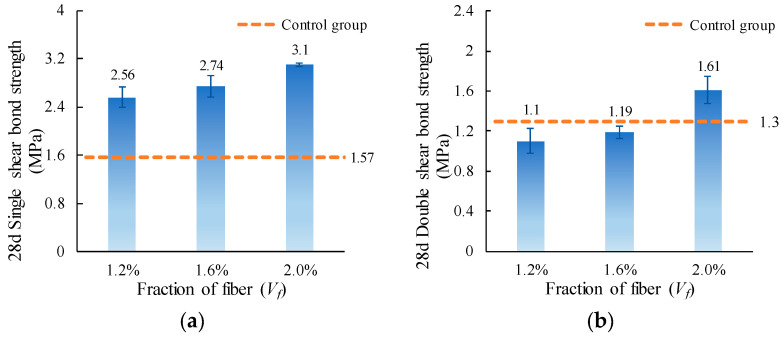
(**a**) Single shear bond strength and (**b**) double shear bond strength of specimens with varying fiber volume fraction (*V*_f_).

**Figure 11 materials-15-04851-f011:**
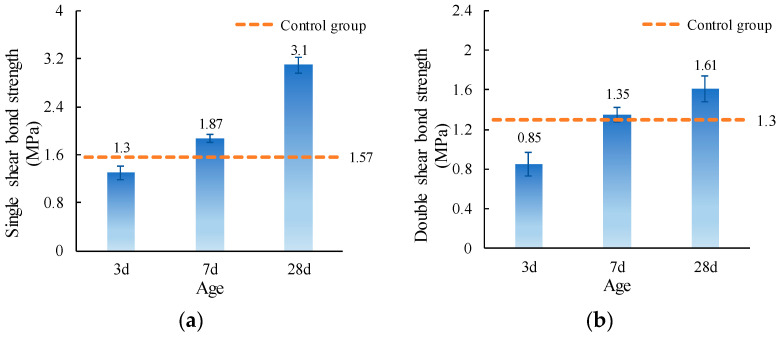
(**a**) Single shear bond strength and (**b**) double shear bond strength of specimens with varying age.

**Figure 12 materials-15-04851-f012:**
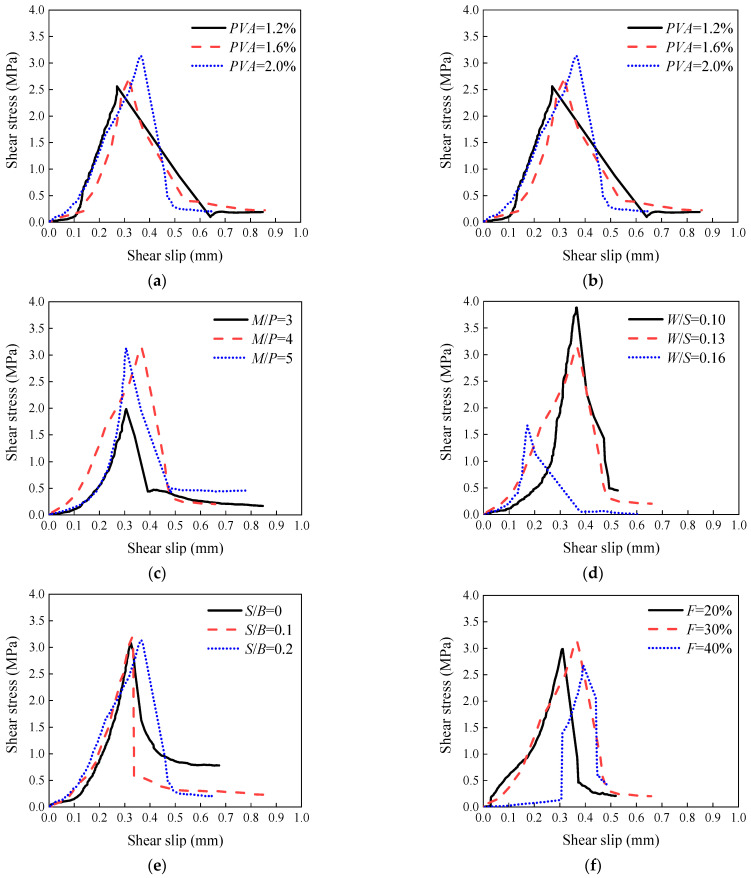
Bond–slip curves under different (**a**) *V_f_*; (**b**) *B*; (**c**) *M*/*P*; (**d**) *W*/*S*; (**e**) *S*/*B*; (**f**) *F*.

**Figure 13 materials-15-04851-f013:**
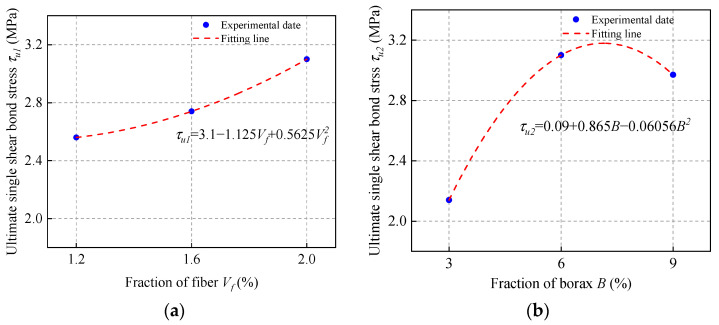
The fitting calculation of (**a**) *τ_u_*_1_; (**b**) *τ_u_*_2_; (**c**) *τ_u_*_3_; (**d**) *τ_u_*_4_; (**e**) *τ_u_*_5_; (**f**) *τ_u_*_6_.

**Figure 14 materials-15-04851-f014:**
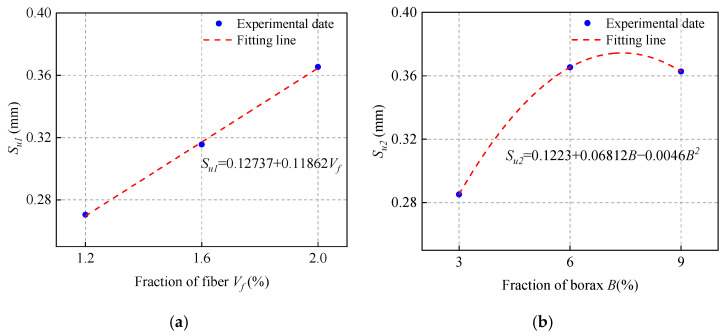
The fitting calculation of (**a**) *S_u_*_1_; (**b**) *S_u_*_2_; (**c**) *S_u_*_3_; (**d**) *S_u_*_4_; (**e**) *S_u_*_5_; (**f**) *S_u_*_6_.

**Figure 15 materials-15-04851-f015:**
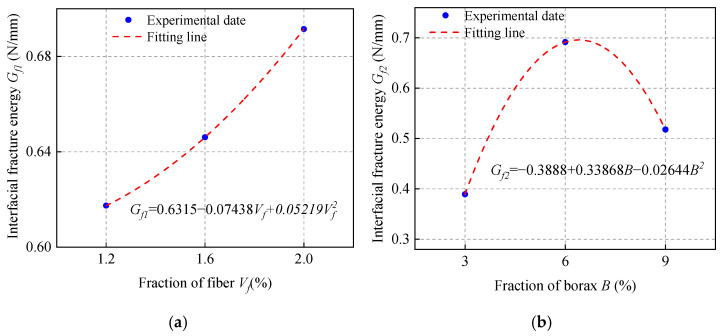
The fitting calculation of (**a**) *G_f_*_1_; (**b**) *G_f_*_2_; (**c**) *G_f_*_3_; (**d**) *G_f_*_4_; (**e**) *G_f_*_5_; (**f**) *G_f_*_6_.

**Table 1 materials-15-04851-t001:** Technical specifications of quartz sand.

SiO_2_ Content	Mohs Hardness	Porosity	Specific Gravity (g/cm^3^)
99.3%	7.5	43%	2.66

**Table 2 materials-15-04851-t002:** Technical specifications of fly ash.

Fineness (45-Micron Standard Square Hole Sieve Allowance)	Water Requirement Ratio	Loss on Ignition	Moisture Content	Sulfur Trioxide	Free Calcium
6%	92%	4%	3%	1.48%	/

**Table 3 materials-15-04851-t003:** Chemical composition of fly ash.

Chemical Composition	SiO_2_	Al_2_O_3_	Fe_2_O_3_	CaO	MgO
Percentage of mass (%)	53.97	31.15	4.16	4.01	1.01

**Table 4 materials-15-04851-t004:** PVA fiber performance indicators.

Diameter (μm)	Length (mm)	Tensile Strength (MPa)	Modulus of Elasticity (GPa)	Elongation at Break (%)	Density (g/cm^3^)
40	12	1560	41	6.5	1.3

**Table 5 materials-15-04851-t005:** Chemical composition of P.O.42.5 ordinary portland cement.

Mineral Composition	C_3_S	C_2_S	C_3_A	C_4_AF	Gypsum
Mass percent (%)	55.5	19.1	6.5	10.1	5

**Table 6 materials-15-04851-t006:** Mineral composition of P.O.42.5 ordinary portland cement.

Mineral Composition	C_3_S	C_2_S	C_3_A	C_4_AF	Gypsum
Mass percent (%)	55.5	19.1	6.5	10.1	5

**Table 7 materials-15-04851-t007:** Mixing proportions.

Mix ID	*W*/*S*	*S*/*B*	*M*/*P*	*V*_f_ (%)	*F* (%)	*B* (%)
*W*/*S*-0.10	0.10					
*W*/*S*-0.13	0.13	0.2	4	2.0%	30%	6%
*W*/*S*-0.16	0.16					
*S*/*B*-0		0				
*S*/*B*-0.1	0.13	0.1	4	2.0%	30%	6%
*S*/*B*-0.2		0.2				
*M*/*P*-3			3		20%	
*M*/*P*-4	0.13	0.2	4	2.0%	30%	6%
*M*/*P*-5			5		40%	
*V*_f_-1.2%				1.2%		
*V*_f_-1.6%	0.13	0.2	4	1.6%	30%	6%
*V*_f_-2.0%				2.0%		
*F*-20%					20%	
*F*-30%	0.13	0.2	4	2.0%	30%	6%
*F*-40%					40%	
*B*-3%						3%
*B*-6%	0.13	0.2	4	2.0%	30%	6%
*B*-9%						9%

## Data Availability

The data presented in this study are available on request from the corresponding author. The data are not publicly available due to privacy.

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
