# Peer review of "Bond Properties of Magnesium Phosphate Cement-Based Engineered Cementitious Composite with Ordinary Concrete"

_materials, 2022, doi:10.3390/ma15144851_

Round 1

Reviewer 1 Report

Comments:
Overall, authors developed an interesting research “Bond properties of magnesium phosphate cement-based engineered cementitious composite with ordinary concrete”.
However, this manuscript should be accepted after a major revision and improvements to eradicate technical, editorial, and presentation weaknesses. Few weaknesses observed are presented below to guide authors for reproducing a better write-up.

1. Could the Authors give explanation of chosen the W/S ratio for tests?

2. Please add information of all technique description (producer, city, country) (2. section).

3. Figure 3 and 4, I think it is not logical to submit the both figure of device and schematic view? This figures do not show any significant results.

4. How many specimens was used for tests? In some sections the authors actually are able to present average results (3 section). It would be helpful to provide each time the number of tested samples.

4. The methodology is not easy to follow and organized.

5. what is P.O.42.5 ordinary portland cement? why need to use this?

6. From the results, it seems that the single shear bond strength of samples is very significant (Fig. 10), while the double shear bond strength increases? It means that the fraction of fiber does not need to be too high?

7. The quality of figures and references list should be improved. A style guide is included at the " Instructions for Authors".
8. Accordingly, the paper may be accepted for publication after major revision.

Author Response

Reviewer #1

Comment: Overall, authors developed an interesting research “Bond properties of magnesium phosphate cement-based engineered cementitious composite with ordinary concrete”. However, this manuscript should be accepted after a major revision and improvements to eradicate technical, editorial, and presentation weaknesses. Few weaknesses observed are presented below to guide authors for reproducing a better write-up.

Reply: The authors appreciate the editor’s valuable comments and suggestions. Corresponding revisions and corrections have been made in the revised manuscript. The detailed replies to each comment are addressed as follows.

Comment 1: Could the Authors give explanation of chosen the W/S ratio for tests?

Reply: In our group's previous research, when the W/S is 0.13, MPC-ECC has good compressive and bending properties. Therefore, the W/S of 0.13 was selected as the benchmark group in this study, and then the effect of W/S on the bond performance was explored.

Comment 2: Please add information of all technique description (producer, city, country) (2. section).

Reply: Thanks for the suggestion. All material technique descriptions are provided by the manufacturer, and all material sources are supplemented and complete in Section 2.1.

Comment 3: Figure 3 and 4, I think it is not logical to submit the both figure of device and schematic view? These figures do not show any significant results.

Reply: Thanks for the suggestion. Now only the pictures of the experimental setup are kept.

Comment 4: How many specimens was used for tests? In some sections the authors actually are able to present average results (3 section). It would be helpful to provide each time the number of tested samples.

Reply: This is described in section 2.3 (2) that “Three specimens were prepared for each mix proportion.”. This has been highlighted in section 2.4.

Comment 5: The methodology is not easy to follow and organized.

Reply: Because there is no unified standard, the test method commonly used for bond performance test is adopted.

Comment 6: what is P.O.42.5 ordinary portland cement? why need to use this?

Reply: P.O.42.5 ordinary portland cement is defined by GB 175-2007, which means that the 28d compressive strength of cement mortar mixed in accordance with the specifications can reach more than 42.5MPa. This is the cement most commonly used in China's construction industry.

Comment 7: From the results, it seems that the single shear bond strength of samples is very significant (Fig. 10), while the double shear bond strength increases? It means that the fraction of fiber does not need to be too high?

Reply: It can be seen from the test that the bond performance of the specimen is the best when Vf is 2%. In our previous study, we found that when the fiber content exceeded 2%, the performance of MPC-ECC itself could be adversely affected. After comprehensive consideration, it is recommended that the fiber content should be 2% in this paper.

Comment 8: The quality of figures and references list should be improved. A style guide is included at the " Instructions for Authors".

Reply: Thanks for the comment. This part has been modified.

Reviewer 2 Report

Dear authors!

Your manuscript contains many weel writen and presented results. But, you should improve manuscript in particular the introduction, methods and conclusions parts. A few comments that I placed as comments with the faile attached.

Author Response

Reviewer #2

The authors appreciate the reviewer’s valuable comments and suggestions. Corresponding revisions and corrections have been made in the revised manuscript. The detailed replies to each comment are addressed as follows.

Comment 1: Please, add the link.

Reply: The link has been added, please see line 38.

Comment 2: It is well known that a high alumina cement and an inorganic polymer binder also have these properties. Why did you not сonsider these materials as repairing OPC materials? What do you think about it? This should be addressed in the introduction.

Reply: At present, there are many new materials being studied as repair materials, some of which have similar properties, but all have their own unique characteristics. Compared with high alumina cement and inorganic polymer binders, the MPC-ECC can quickly obtain early strength so that the repaired components can be restored to use as soon as possible. This is an obvious feature. Of course, in the future, we will also carry out relevant studies on other materials to make a more detailed comparison.

Comment 3: The sum of oxides are 94,38 only. What is the rest?

Reply: The rest are some minorities including Na2O, f-CaO, Loss, and Cl- and have been supplemented. Please see Table 5.

Comment 4: You should add the diffractogram and physical properties (compressive strength, etc.) of OPC.

Reply: Thanks for the reminding. We are really sorry that the diffractogram of cement is not available now, but we will keep that in mind and try to provide such information in the future research. The physical properties of OPC have been added, please see line 160-162.

Comment 5: This text has a wrong format.

Reply: This text has been modified, please see line 198.

Comment 6: It would be better if you add age of binders in the figure sign. For other figures also.

Reply: Thanks for the suggestion. All relevant figures have been modified. Please see Figure 5-10.

Comment 7: Why did you study influence of age for MPC-ECC binders without other additions? You should indicate this. How is influence other additions on binder properties for other ages?

Reply: Thanks for the suggestion. This part is mainly to explore the early strength performance of MPC-ECC, and its bond strength can reach the level of the control group within 7 days.

Comment 8: You need to add the link to the Bound-slip modeling method described.

Reply: The link has been added, please see line 448.

Comment 9: I think it is better to use the bigger font and below for symbol also.

Reply: Thanks for reminding. The font has been modified. Please see line 450-453.

Comment 10: These figures is very small.

Reply: Thanks for the suggestion. The figure size has been modified. Please see Figure 12-15.

Comment 11: What is the optimal composition of binder according your model and experiment? What do you think about economical effectively by using your binder for repair OPC compares to other materials?

Reply: “The bond properties of MPC-ECC are better when the W/S is 0.10, M/P is 5, B is 6%, Vf is 2%, F is in the range of 20%-30%, and the S/B has little effect on its bond performance.” This part has been added to the conclusion. From the perspective of economy, MPC-ECC has no obvious advantages over ordinary concrete. However, under specific conditions, such as emergency maintenance of roads and runway maintenance of airports, that need to be completed as soon as possible, the use of fast hardening and early strength repair materials such as MPC-ECC can solve practical problems faster, thus reducing economic losses.

Round 2

Reviewer 1 Report

accept in present form